# Metabolic and Body Composition Changes in Ice Hockey Players Using an Ergogenic Drug (Cytoflavin)

**DOI:** 10.3390/biology12020214

**Published:** 2023-01-29

**Authors:** Victoria Zaborova, Vladislav Kurshev, Kira Kryuchkova, Valeria Anokhina, Vladimir Malakhovskiy, Vera Morozova, Veronika Sysoeva, Giovanna Zimatore, Valerio Bonavolontà, Laura Guidetti, Yuliya Dronina, Elena Kravtsova, Dmitry Shestakov, Konstantin Gurevich, Katie M. Heinrich

**Affiliations:** 1Institute of Clinical Medicine, Sechenov First Moscow State Medical University, 119991 Moscow, Russia; 2Laboratory of Sports Adaptology, Moscow Institute of Physics and Technology (National Research University), 141700 Dolgoprudny, Russia; 3Department of Theoretical and Applied Sciences, eCampus University, 22060 Novedrate, Italy; 4IMM-CNR, Institute for Microelectronics and Microsystems, 40129 Bologna, Italy; 5Department of Biotechnological and Applied Clinical Sciences University of L’Aquila, 67100 L’Aquila, Italy; 6Unicusano Department, Università Niccolò Cusano, 00166 Rome, Italy; 7Institute of Public Health, Sechenov First Moscow State Medical University, 119991 Moscow, Russia; 8Moscow Clinical Scientific Center named after A. S. Loginov, 111123 Moscow, Russia; 9UNESCO Chair, A.I. Yevdokimov Moscow State University of Medicine and Dentistry, 127473 Moscow, Russia; 10Department of Public Health, Research Institute of Healthcare Organization and Medical Management of Moscow Department of Healthcare, 115184 Moscow, Russia; 11Department of Kinesiology, Kansas State University, Manhattan, KS 66506, USA

**Keywords:** ice hockey players, Cytoflavin, body composition, bioelectrical impedance analysis, aerobic fitness, overtraining

## Abstract

**Simple Summary:**

Performance in sports is determined by technical, tactical, psychological, and social factors and physiological characteristics of an athlete. Hockey players’ anthropometric indicators such as the ideal level of muscle mass and body fat of athletes correlate with their performance. In team sports, the use of ergogenic drugs for recovery is relevant to avoid athletes’ overtraining. The aim of the work was to assess the effect of Cytoflavin consumption on body composition and aerobic performance in professional ice hockey players. The results of the study showed that Cytoflavin exerts a positive effect on performance parameters. Our research can contribute to changing the approach to improving and maintaining the effectiveness of sports training and achieving maximum per-formance in physical activity.

**Abstract:**

Background and objectives: In ice hockey, the major physical workload comes from acceleration in all planes of motion and transitions between skating trajectories. Hockey players’ anthropometric characteristics correlate with performance. In team sports, the use of ergogenic drugs for recovery is relevant to avoid athletes’ overtraining. It is very important to protect athletes’ health and allow them to maintain high-performance levels. Cytoflavin is an ergogenic drug whose action is based on the combined effects of its active ingredients (succinic acid, inosine, nicotinamide and riboflavin), which are naturally occurring metabolites that stimulate tissue respiration. The study aimed to assess the 6-week Cytoflavin consumption effects on body composition (body weight, body mass index, body fat percentage and bioimpedance phase angle) and aerobic performance. Methods: This study included 60 male professional hockey players (aged 19 to 36 years) divided into two groups of 30 subjects: group I (body weight 87.90 ± 7.44 kg, BMI 25.86 ± 2.04 kg/m^2^) and group II (body weight 87.04 ± 6.22 kg, BMI 25.52 ± 2.38 kg/m^2^). Athletes in group I received Cytoflavin, whereas athletes in group II did not. Results: In group I, statistically significant reductions in body weight and body mass index were not observed until 14 and 35 days, respectively. In contrast, in group II, both body weight and BMI significantly decreased both times. Aerobic performance significantly increased in both groups, with significantly greater increases in group I. Conclusions: Cytoflavin can be considered an ergogenic drug that improves body composition parameters, especially in the control of weight reduction and improvement in aerobic performance.

## 1. Introduction

Ice hockey is a physiologically demanding sport that requires aerobic and anaerobic metabolism as energy sources [1]. The aerobic capacity of hockey players increases as they grow and mature physically and physiologically [2,3]. The major physical workload comes from acceleration in all planes of motion, and accumulation of workload occurs mainly due to running and other locomotor movements [4]. Ice hockey requires rapid transitions between skating trajectories to move effectively on the ice surface. Player performance is largely related to effective change-of-direction maneuvers and inherent asymmetric dynamic behaviors [5]. 

Peterson et al. [6] point out that the anthropometric and physiological characteristics of ice hockey players are correlated with performance. The majority of athletes practicing winter sports go through periods of highly intensive training. Therefore, they need higher energy and nutrient intake as well as sufficient nutrition and hydration before, during and after training [7].

Having ideal levels of muscle mass and body fat and meeting nutritional needs during training are the main challenges for ice hockey team players; indeed, Vigh-Larsen et al. [8] suggested the importance of on-ice high-intensity training in professional players in addition to training aimed to develop lean body mass in young athletes. In team sports, recovery between games is one of the top priorities related to rehydration, restoration and adaptation [9]. Such challenges give rise to a physiological need to use allowed ergogenic aids for the correction of metabolic deficiencies that can enhance and shorten the time of adaptive responses in order to progressively increase training.

Cytoflavin produced by NTFF Polisan (Scientific and Technological Pharmaceutical Firm POLYSAN LLC, 192102, St. Petersburg; https://eng.polysan.ru, accessed on 16 July 2022) is a drug whose active principles are succinic acid, inosine, nicotinamide and riboflavin; it does not contain substances prohibited for use in sports [10]. Each Cytoflavin tablet contains 300 mg of succinic acid, 50 mg of inosine (riboxinum), 25 mg of nicotinamide, 5 mg of riboflavin mononucleotide, and the following excipients: medium molecular weight povidone, calcium stearate, methacrylic acid and ethyl acrylate copolymer, propylene glycol, azorubine and tropaeolin-O. Each vial of succinic acid (10 mL) solution for intravenous administration contains 1 g of succinic acid, 200 mg of inosine (riboxinum), 100 mg of nicotinamide, 20 mg of riboflavin mononucleotide, and the following excipients: N-methylglucamine, sodium hydroxide, water for injection and meglumine (N-methylglucamine).

The pharmacological action of Cytoflavin is based on the combination of effects from its ingredients, which are naturally occurring metabolites that stimulate tissue respiration [11]. Succinic acid is an endogenous intracellular metabolite of the Krebs cycle that performs a universal function of energy production in the cells. Succinic acid is rapidly converted into fumaric acid by the mitochondrial enzyme succinate dehydrogenase and coenzyme flavin adenine dinucleotide (FAD) and is then converted into other metabolites of the citric acid cycle. Succinic acid enhances aerobic glycolysis and adenosine triphosphate (ATP) production in the cells. Succinic acid improves cellular respiration by promoting electron transport in mitochondria. Riboflavin (vitamin B_2_) is a FAD coenzyme that activates succinate dehydrogenase and other oxidation–reduction reactions of the Krebs cycle. In cells, nicotinamide (vitamin PP) is converted, through a cascade of biochemical reactions, into nicotinamide adenine dinucleotide (NAD) and nicotinamide adenine dinucleotide phosphate (NADP), catalyzing nicotinamide-dependent enzymes of the Krebs cycle that are necessary for cellular respiration and activation of ATP production. Inosine is a purine derivative, an ATP precursor. It can activate several Krebs cycle enzymes promoting the production of key nucleotide enzymes: FAD and NAD.

In summary, according to Coffey [12], the molecular response to increased exercise intensity consists of the biochemical signals that activate adaptive cellular pathways and include (i) an increase in the ratio of adenosine monophosphate (AMP) to ATP, (ii) an increase in the levels of reactive species of oxygen and nitrogen, (iii) a depletion of muscle glycogen levels and (iv) a decrease in oxygen tension [12,13,14]. In this context, Cytoflavin enhances cellular respiration and energy production, improves oxygen uptake in the tissues, restores the activity of enzymes with an antioxidant effect, activates intracellular protein production and promotes glucose and fatty acid uptake and neuronal GABA re-synthesis through the GABA shunt [15].

All these processes are involved in the aerobic pathway for energy production, which occurs during physical activities. Indeed, the maximal oxygen uptake (VO_2_ max) during dynamic muscular exercise is commonly taken as a crucial determinant of the ability to sustain high-intensity exercise [16]. Cytoflavin was recently used in human studies on hemorrhagic stroke [17], where, by the study of the parameters of oxidative stress and routine biochemistry blood test, it was observed that Cytoflavin treatment allowed for a stabilized clinical–laboratory picture of hemorrhagic stroke, improved treatment results and reduced hospital mortality rates. Moreover, Silina et al. [17] reported that Cytoflavin contains coenzymes nicotinamide, riboflavin and inosine (riboxin), which stimulate glucose–pyruvate transformations in the Krebs cycle. In summary, Cytoflavin use has been studied in the medicine of critical states in humans and rats, especially in cardiology [17], during ischemia [11] and in spinal cord injury [18], for its antioxidant and antihypoxic effects.

Yet, there are few studies that have investigated the effect of Cytoflavin on athletes. Kosinets et al. [19] reported that Cytoflavin promoted athletes’ physical fitness by improving energy supply, psycho-emotional conditions and competition form, and recommended its use in the pre-competition period. To our knowledge, no studies have analyzed the effects of Cytoflavin on athletes’ body composition and performance parameters, such as aerobic power. Therefore, the aim of our study was to assess the effect of 6-week Cytoflavin consumption on body composition (body weight, body mass index [BMI], body fat percentage and bioimpedance phase angle) and aerobic performance in professional ice hockey players.

## 2. Materials and Methods

The study included 60 male professional hockey players aged 19 to 36 years (mean age 24.1 ± 8.75 years) from Spartak Hockey Club of the Kontinental Hockey League of Russia. The participants were divided into two groups of 30 subjects each: group I (body weight 87.90 ± 7.44 kg, BMI 25.86 ± 2.04 kg/m^2^) and group II (body weight 87.04 ± 6.22 kg, BMI 25.52 ± 2.38 kg/m^2^). No initial statistical differences were found between groups (*p* <0.05). The experimental period included 3 cycles of 12 days of training, with a break between cycles of 5 days for a total of 36 days of training. The first training cycle was focused mainly on speed and strength training. The second training cycle consisted of strength training and technical training on the ice. The third training cycle consisted of technical training on ice and participation in hockey tournaments. Training sessions were held 2 times a day for all cycles, each lasting 1.5 hours.

The two groups of athletes were comparable in age, as well as anthropometric, clinical and instrumental indicators. The following were the criteria for inclusion in the study: male; aged from 19 to 36 years; level of sportsmanship—Candidate for Master of Sports of Russia, Master of Sports of Russia or Master of Sports of Russia of International Class; and presence of signed informed consent. The following were the criteria for not being included in the study: age less than 19 or more than 36 years and gastrointestinal diseases in the acute stage. The following were the exclusion criteria: withdrawal of the athlete from the study protocol and the presence of adverse events.

During the study, group I and group II participants received basic sports nutrition, which included the use of L-carnitine 1500 mg; isotonic drinks, multivitamins and amino acids during training; and protein mixtures as well as gainers after training. In the period that the athletes did not change their eating habits, they did not reduce their competitive activity and they had no injuries.

During the experimental period, group I athletes received Cytoflavin as follows: 10.0 mL of Cytoflavin in 100.0 mL of 5% glucose solution by intravenous drip for 10 days and then 2 tablets twice daily at 8- to 10-h dosing intervals 30 min prior to meals, swallowing tablets without chewing, with sweet tea for 25 days. Group II did not receive any drugs containing succinic acid or other metabolic agents.

This study was approved by the Institutional Ethics Committee of the Federal Scientific Center of Physical Culture and Sports, Russian Federation (protocol number 14 dated 25 November 2019). All subjects included in this study were volunteers according to the Institutional Review Board protocol, in accordance with the principles of the Declaration of Helsinki, and all provided written informed consent. Bioelectrical impedance analysis (BIA) was conducted before the study, on day 14 and on day 35 on the ABC-01 Medass Analyser (Medas Ltd, Moscow, Russia). Medas measurements were obtained in the supine position according to the conventional wrist-to-ankle 4-electrode scheme at an electric current frequency of 50 kHz with the disposable surface bio-adhesive Ag-AgCl ECG electrodes (F3001ECG, FIAB SpA, Vicchio, Italy) placed on the right side of the body [20]. Anthropometry was performed according to a standardized protocol [21]. Body weight, BMI, body fat percentage and bioimpedance phase angle were measured. Standing height was measured to the nearest 0.1 cm without shoes and socks using a Seca stadiometer (Model 217, seca gmbh & co. kg, Hamburg, Germany). Weight was measured to the nearest 0.1 kg in underwear using a mechanical column scale with eye-level beam Seca (Model 700, seca gmbh & co. kg, Germany). BMI was calculated as the ratio weight/height^2^ (kg/m^2^). According to the standardized scale of phase angle normal values, phase angle values were categorized as low (less than 4.4), decreased (4.4 to 5.4), normal (5.4 to 7.8) and very high (more than 7.8).

Moreover, as evaluation of cardiorespiratory parameters is important in athletic populations in order to monitor their training status, an ergo-spirography analysis was conducted to evaluate the participants’ maximal aerobic power (VO_2max_) during an exhaustive incremental exercise test. Prior to the beginning of training and after the last training session (after 35 days), cardiopulmonary incremental exercise testing (CPET) was performed. The procedure for the incremental exercise test CPET on a treadmill (MTM-1500, Schiller, Germany) was as follows: after a 1-min rest period while stated on the treadmill, the active phase started consisting of an initial speed of walking at 5 km/h followed by speed increments of 2 km/h every 2 min without changing the slope until voluntary exhaustion or attainment of VO_2max_. 

Specifically, VO_2max_ was identified when oxygen uptake reached a plateau [16] or started to fall even though the work rate kept increasing. Secondary criteria were also applied to verify the maximal effort, such as attainment of age-predicted maximum heart rate (maximal age-specific heart rate is reckoned as 220-age) and/or the respiratory exchange ratio greater than 1.10 [1]. 

The heart rate (HR) was recorded by a chest belt (Polar electro OY, Kempele, Finland) simultaneously with measurements of oxygen consumption (VO_2_, mL/min), carbon dioxide production (VCO_2_, mL/min) and pulmonary ventilation (VE, mL/min) that were measured by an automatic gas analyzer (Quark PFT Cosmed™, Pomezia, Italy).

### Statistical Analysis

Means and standard deviations for body weight, BMI, body fat mass and phase angle are shown in Tables 1–3. To detect significant differences in the measured variables between groups at different time points, a repeated-measures ANOVA was applied. When statistically significant differences were found, a Tukey test for paired samples was used as a post hoc analysis. Statistical significance was set at *p* ≤ 0.05. Microsoft Excel for Windows and STATGRAPHICS 18.1.08 (Statpoint Technologies, Inc., The Plains, VA, USA) were used for data analysis.

## 3. Results

### 3.1. Anthropometric Measurements and BIA

Assessment of trends in anthropometric measurements and BIA data collected from athletes at different time intervals are presented by groups (Table 1, Figure 1 and Figure 2).

Assessment of trends in the body weight and BMI values showed that changes occurred in both group I and group II throughout the study. However, changes in body weight and BMI values occurred differently in the groups. Specifically, in group II, there was a significant reduction in studied parameters as compared to the baseline throughout the study, whereas in group I, statistically significant reductions in body weight and BMI were not observed until day 14 (*p* <0.001 and *p* <0.021, respectively) followed by a stabilization of these parameters on day 35 as compared to baseline (*p* = 0.092 and *p* = 0.06, respectively).

Assessment of the trends in the BIA values showed a statistically significant increase in the impedance phase angle in both groups throughout the study with mean values of 7.43° ± 0.28º in group I and 7.33° ± 0.31º in group II (*p* < 0.001 as compared to the baseline) on day 35 (Table 2, Figure 3 and Figure 4).

Body fat values significantly decreased on day 14 both in group I and group II as compared to the baseline (*p* < 0.001), whereas on day 35 there was no significant difference as compared to the baseline in both groups (*p* = 0.873 and *p* = 0.097, respectively). At the same time, BIA values did not show any statistically significant differences between the groups (*p* > 0.05) at any time point.

### 3.2. Maximal Oxygen Uptake (VO_2_ Max) and Time to Exhaustion

The indicators of maximum oxygen consumption (VO_2_ max) and time to exhaustion on the 35th day of the study in group I were statistically significantly higher than those in group II (*p* < 0.05). In group I, during the study, the average VO_2_ max increased by 11.2%, while the increase in time to exhaustion was 3.9%. VO_2_ max values for the groups averaged 52.25 ± 4.2 mL/kg/min and 49.06 mL/kg/min (*p* = 0.003) (Table 3).

Time to exhaustion in group I was also significantly higher than in group II. The time to exhaustion indicators increased in group I by an average of 12.9%, in group II by 10.8% and averaged 14.15 ± 1.30 min and 13.27 ± 1.04 min (*p* < 0.05), respectively. The median increase for this indicator was 14.8% in group I and 11.6% in group II.

## 4. Discussion

Recovery in athletes is a crucial part of the training process and it is then vital to, first, protect the health of athletes and, second, allow them the ability to maintain optimal performance levels. The aims of our study were to assess the effects of Cytoflavin consumption on body composition parameters and performance in hockey players. In this work, body composition parameters (body weight, BMI, body fat mass percentage and BIA phase angle) were measured and compared before and after the 6 weeks of Cytoflavin consumption (group I) and compared with a control condition (group II). Results showed that on day 35 of the preparatory period, there was a significant decrease in body weight and BMI in the professional hockey players who did not receive treatment with Cytoflavin (group II).

Moreover, the loss of body weight observed in group II can be attributed to a loss in muscle mass as the percent of fat mass was not significantly changed on day 35; on the contrary, the BIA parameter phase angle showed that the hydration status changed in both groups (Table 1 and Table 2). These results are relevant for ice hockey team players in order to control ideal levels of muscle mass and body fat. Specifically, sequential treatment with succinic acid helped to stabilize BIA values, which suggests an improved adaptation to physical workload in professional hockey players during the preparatory period.

On the basis of current knowledge, during the training sessions in athletes, it is crucial to monitor changes in lean and fat mass, separately, to preserve agility (due to a body mass increase) and to improve muscle strength (lean mass increase) [22].

Cytoflavin can be considered an ergogenic drug that improves body composition parameters, especially controlling weight reduction and adaptation to physical workload as previously reported by Kosinets [19]. Furthermore, referring to measured metabolic parameters (VO_2_ max and time to exhaustion), a statistically significant increase in maximal oxygen consumption (VO_2_ max) and an increase in time to exhaustion during the incremental test were found in group I compared with group II, indicating a positive adaptation to physical workload for those subjects who received Cytoflavin.

Thus, Cytoflavin exerts a positive effect on performance parameters. Further studies are necessary to investigate deeper on this topic. Indeed, while Kosinets [19] concluded that succinate-containing preparations promoted physical fitness, Brown reported that succinate and other tricarboxylic acid cycle intermediates did not improve physical performance [23]. Future extensions of this work should consider the discrepancy between training intensity and the intensity of physical activity in a game situation [1]. In fact, the metabolism of the muscles activated in a game situation is different from that during a cycling ergometer test, which is a continuous activity performed with steady velocity [1]. From a physiological perspective, many differences may occur between an incremental test in laboratory conditions and a game situation, because ice hockey games require athletes to perform frequent accelerations, stops and changes in skating direction [24,25].

To simulate the demands of the game of ice hockey, Stevees et al. [26] proposed specific on-ice game simulations, yet the authors did not find a relationship between maximal aerobic power and recovery during a simulated game in elite hockey players. Recently, a cross-sectional study based on data collected from teams from the best and second-best Danish ice hockey divisions was designed by Vigh-Larsen and co-workers [8], in order to evaluate the fitness profiles of elite and sub-elite male ice hockey players using both off-ice (i.e. measures of height, body mass, body composition and a vertical countermovement jump test) and on-ice testing (i.e. agility performance and sprint ability Yo-Yo IR1-IHSUB were tested, in full-hockey gear and with the stick in hand) procedures with a large sample size [8,27].

Because hockey requires a high aerobic capacity with a sufficient recovery rate, testing this fitness is imperative. Currently, shuttle running tests for determining the aerobic performance of players, performed to failure, have become the most popular in game sports. Test performance can be equated to performance during the most intense game periods, where slow recovery can be a hindrance. These tests are related to maximum oxygen consumption (MPC or VO_2max_). The maximum shuttle test (peak test) is characterized by a greater load, expressed in acceleration and deceleration, which leads to the accumulation of a high lactate concentration and slow recovery [28]. The choice of a test for aerobic abilities should be aimed at minimizing injury risks while preserving health. In the 7 × 50 m test, players perform a larger amount of running at high speed, which leads to a significant increase in lactate concentration, including within five minutes after the test is completed. The interval shuttle run test is characterized by a lower load and lactate concentration after the test and during the recovery period compared with the maximum [29]. Recent publications report on generally accepted indicators of the characteristics of functional fitness of football players of various roles. The average values for maximum oxygen consumption (MPC), yo-yo test, maximum heart rate, the result of running 30 m, and jumping from squat/high jump vary depending on the nationality of the players, competitive level and role. On average, these players achieve MPC indicators in the range from 45.1 to 55.5 mL/kg/min, and the results of the yo-yo test are 780–1379 m, the maximum value of the heart rate is 189-202 beats per minute, the time of the 30 m run is 4.34–4.96 s and the results of the vertical jump from the squat are 28–50 see [30]. In comparison, our study participants averaged a VO_2_ max of 49–52 mL/kg/min. In studies aimed at studying the level of aerobic fitness of basketball players with hearing impairment, ergospirometry (on a bicycle ergometer) was used. In particular, the peak oxygen consumption (VO2peak) was measured. At the same time, the best cardiorespiratory fitness of deaf athletes correlated with a higher skeletal muscle mass and a lower percentage of fat mass [31]. All these recent studies have demonstrated the importance of studying the specific fitness characteristics of every sports game to safely improve the performance of athletes. Furthermore, recently, HR time series were recorded during physical exercise to study athletes and monitor their health status, to assess their fitness level [32,33] and to prevent injuries. HR variability and the other related parameters could enrich the knowledge in future studies.

## 5. Study Limitations

This study was limited to the sport studied (hockey), a small sample size of 60 participants and the 35-day observation period. Yet, we had high participant adherence to the study protocol and study participants were highly qualified athletes, including 25 who were Masters of Sports of International Class (MSIC), 26 athletes who were Masters of Sports (MS) and 9 who were candidates for Masters of Sports (CMS). Future research with larger samples and longer study periods may help confirm these results. Further research is needed to study the effects of Cytoflavin on athletes of other disciplines.

## 6. Conclusions

Therefore, group I athletes received 10.0 mL of Cytoflavin in 100.0 mL of 5% glucose solution by intravenous drip for 10 days and then 2 tablets twice daily for 25 days. In the main group (group I), there was no statistically significant reduction in body weight and body mass index compared with the control group (group II), in which both body weight and BMI decreased significantly. Aerobic performance significantly increased in both groups, with significantly greater increases in group I. In conclusion, the present work suggests that Cytoflavin can play a positive role in both the maintenance of lean body mass in off-season training and the improvement in parameters linked to aerobic performance in professional ice hockey players. The research can contribute to changing the approach to improving and maintaining the effectiveness of sports training and achieving maximum performance in physical activity.

## Figures and Tables

**Figure 1 biology-12-00214-f001:**
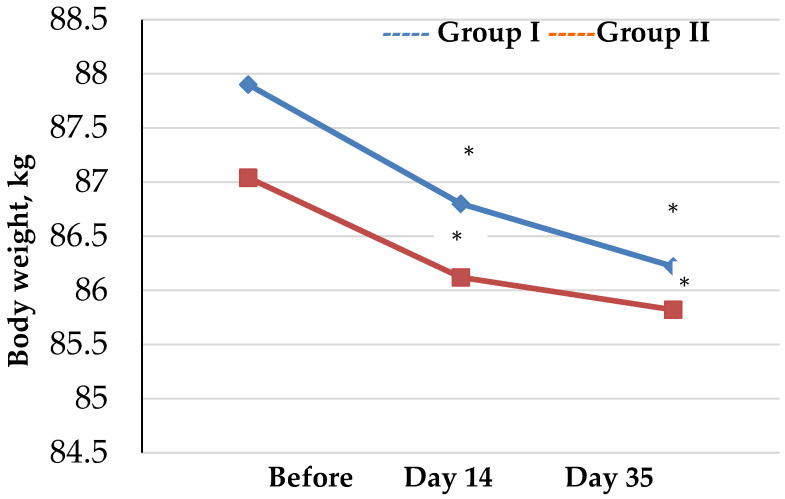
Trends in body weight values of athletes by group (* *p* < 0.05—statistical significance of differences within each group compared to the baseline).

**Figure 2 biology-12-00214-f002:**
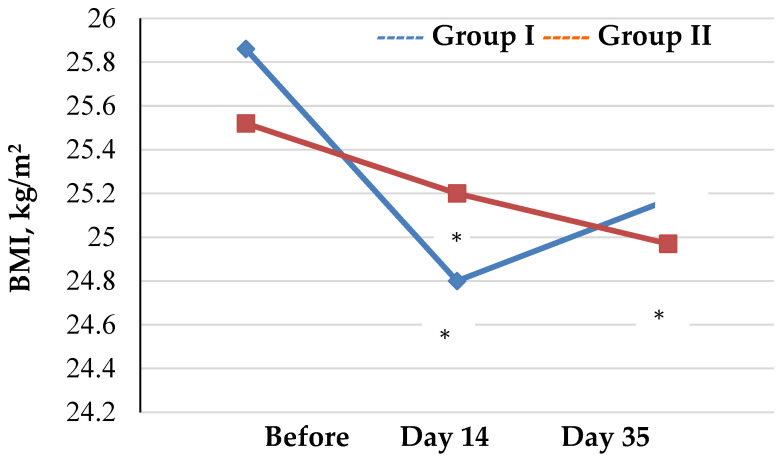
Trends in BMI values of athletes by group (* *p* < 0.05—statistical significance of differences compared to the baseline).

**Figure 3 biology-12-00214-f003:**
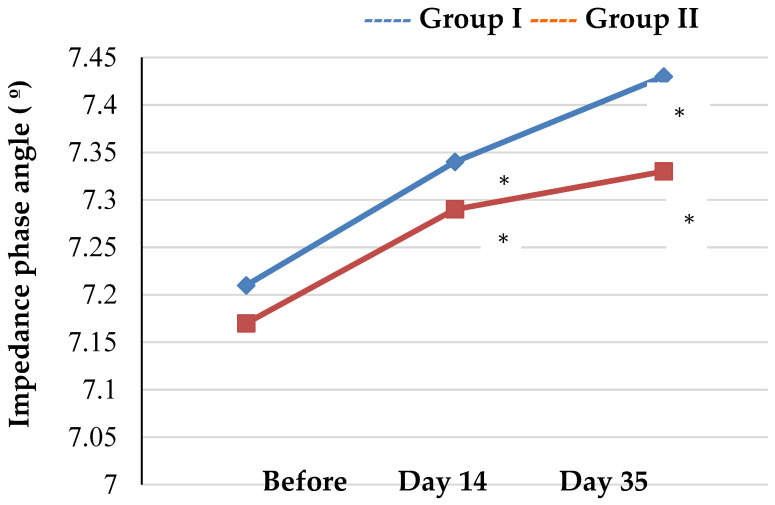
Trends in impedance phase angle of athletes by groups (* *p* < 0.05—statistical significance of differences compared to the baseline).

**Figure 4 biology-12-00214-f004:**
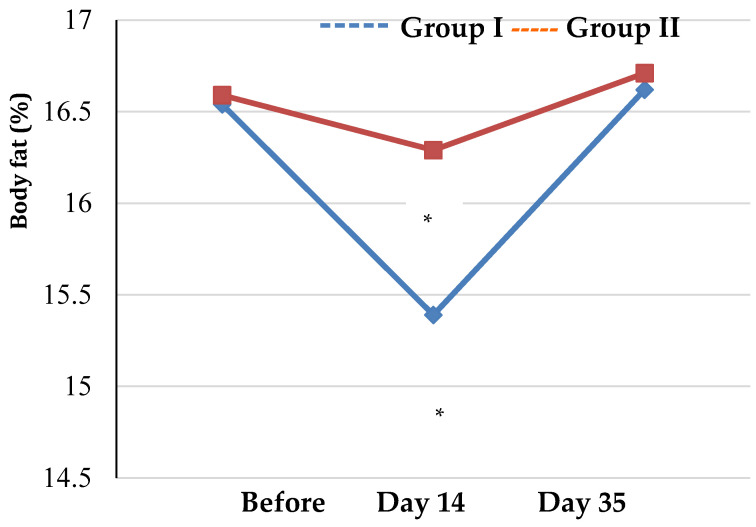
Trends in body fat values of athletes by group (* *p* < 0.05—statistical significance of differences compared to the baseline).

**Table 1 biology-12-00214-t001:** Trends in anthropometric parameters of athletes by groups.

Parameter	Time Point	Group I	Group II
Body weight (kg)	Pre-study	87.90 ± 7.44	87.04 ± 6.22
	Day 14	86.80 ± 7.16″	86.12 ± 6.72′
	Day 35	86.22 ± 6.74	85.82 ± 7.14′
BMI (kg/m^2^)	Pre-study	25.86 ± 2.04	25.52 ± 2.38
	Day 14	24.80 ± 2.27′	25.20 ± 2.3″
	Day 35	25.17 ± 2.19	24.97 ± 2.11′

′*p* < 0.05, ″*p* < 0.001—statistical significance of differences compared to the baseline.

**Table 2 biology-12-00214-t002:** Trends in BIA parameters of athletes by group.

Parameter	Time Point	Group 1	Group 2
Body fat (%)	Pre-study	16.54 ± 3.7	16.59 ± 2.83
	Day 14	15.39 ± 3.36′	16.29 ± 2.93′
	Day 35	16.62 ± 2.81	16.71 ± 2.82
Impedance phase angle (°)	Pre-study	7.21 ± 0.43	7.17 ± 0.36
	Day 14	7.34 ± 0.34′	7.29 ± 0.34′
	Day 35	7.43 ± 0.28′	7.33 ± 0.31′

′*p* < 0.001—statistical significance of differences compared to the baseline.

**Table 3 biology-12-00214-t003:** Metabolic parameter during exhaustion incremental test by group.

Indicators	Period	Group 1	Group 2
Time to exhaustion (min)	Before the study	12.53 ± 1.40	11.98 ± 0.9
	Day 35	14.15 ± 1.30	13.27 ± 1.04′
	Significance level, *p*	< 0.001	< 0.001
VO_2 max_ (mL/kg/min)	Before the study	46.99 ± 3.17	47.21 ± 2.84
	Day 35	52.25 ± 4.20	49.06 ± 2.76′
	Significance level, *p*	<0.001	<0.001

′*p* < 0.05—significance of differences compared with group I.

## Data Availability

The relevant data generated and/or analyzed in the current study are available from the corresponding author upon reasonable request.

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
