# Peer review of "Metabolic and Body Composition Changes in Ice Hockey Players Using an Ergogenic Drug (Cytoflavin)"

_biology, 2023, doi:10.3390/biology12020214_

Round 1
Reviewer 1 Report
The authors take into consideration the use of an ergogenic supplement in Hockey, I report some aspects that can implement the quality of the presentation of the results obtained
Line 65
Indicate the references of the manufacturing company (address, etc.) in addition to the website
Line 122
Specify the championship or federation to which the athletes belong
Line 125
<0.05
Line 136-137 specify region/country of the Federal Scientific Center of Physical Culture and Sports
Line 142
Specify references of the BIA analyzer manufacturer, specify which electrodes were used to perform the BIA detection, specify the operating protocol used in the detection of the bia e.g.: https://doi.org/10.3390/jfmk7040086 par 2.2 -2.3
Specify if software was used to obtain the percentage of body fat or if the data was directly supplied by the machine
Line 142
Specify the instrumentation used to collect the anthropometric data (Name, manufacturing company, references)
Line 148 -152
Specify in more detail the operative protocol used to perform exhaustive incremental exercise tests
Line 150
Specify the operating protocol used to perform CPET
Line 162
Specify references of the Excell manufacturer
In the materials and methods section, it would be appropriate to specify if in the period followed the athletes changed their eating habits, had injuries or reduced their competitive activity
In the discussion it would be appropriate to deepen the results relating to fat mass, discussing the differences found between T14 and T35 on the basis of current knowledge,
Study Considerations
It would be appropriate to hypothesize which parameters/evaluation could enrich the knowledge in future studies
Conclusions
It would be useful to briefly reiterate the dosage of the product used, the period of intake, which parameters improve and the need for future research
Reviewer 2 Report
Dear,
Manuscript Number: biology- 2125425
Title Manuscript: Metabolic and body composition changes in ice hockey players using an ergogenic drug
This study investigated the effect of 6 weeks of Cytoflavin consumption on body composition parameters (body weight, body mass index, body fat percentage and bioimpedance phase angle) and aerobic performance in ice hockey players. This study is an important and interesting topic since the study participants are professional ice hockey players (professional athletes) but at the moment MAJOR REVISIONS are necessary in order to make it suitable for a final decision for “Biology”.
POINTs of STRENGTH:
1) The effect of Cytoflavin on body composition and aerobic performance in professional ice hockey players;
2) Arguments in the “Discussion” section and content arrangement in the “Introduction section”;
POINTs of WEAKNESS (and/or should be revised to improve the manuscript):
Main title
3) Please add “Cytoflavin” word in the main title of the manuscript as follows:
Metabolic and body composition changes in ice hockey players using an ergogenic drug (Cytoflavin);
Abstract:
4) Please organize the “Abstract” section-unstructured as follows: Background and objectives, Methods, Results, Conclusions;
5) Please specify the classification of groups with their number in the “Abstract/methods” section;
6) Please modify the “keywords” section as follows: Ice hockey players; Cytoflavin; Body composition; Bioelectrical impedance analysis; Aerobic fitness; Overtraining;
1. Introduction:
7) The “introduction” section is one of the POINTs of STRENGTH in this manuscript;
2. Materials and methods
8) The recruitment/screening process of ice hockey players as well as inclusion and exclusion criteria should be described in more detail such as sample size, age, BMI, professional ice hockey players, physical fitness level or VO2max, healthy status, blood pressure, free of medication before starting the study and so on;
9) Please provide a valid reference for the study protocol;
10) Considering the important effect of nutrition on body composition, did the authors monitor the nutritional status of ice hockey players during 6 wees? IF YES, please add the method of controlling nutritional status;
3. Results
11) Please specify Group I and Group II in Tables & Figures;
12) Were side effects of Cytoflavin reported by ice hockey players? IF YES, please provide;
4. Discussion
13) Cytoflavin consumption? OR Cytoflavin assumption? Which is true? Please clarify and correct in the “discussion” section;
14) It seems the authors will agree that the “limitations” section should be added to this study;
5. Conclusions
15) In this study, participants were professional ice hockey players… OR Elite ice hockey players… Which is true? Please clarify;
16) What does this study add to the literature? Please explain and add in the “conclusions” section;
References
17) “References” section is not always in accordance with the authors' guidelines. In particular, please check No. 10 and 27 for validation.
Best Regards
10 January 2023
Round 2
Reviewer 2 Report
Dear Authors,
Manuscript Number: biology- 2125425
Title Manuscript: Metabolic and body composition changes in ice hockey players using an ergogenic drug (Cytoflavin)
I am very grateful to the authors for their efforts.
In general, this manuscript has found suitable content after correcting major revisions, and the modified revisions are accepted.
Best Regards
21 January 2023